# Maternal Distress during Pregnancy and the Postpartum Period: Underlying Mechanisms and Child’s Developmental Outcomes—A Narrative Review

**DOI:** 10.3390/ijms232213932

**Published:** 2022-11-11

**Authors:** Ljiljana Jeličić, Aleksandra Veselinović, Milica Ćirović, Vladimir Jakovljević, Saša Raičević, Miško Subotić

**Affiliations:** 1Cognitive Neuroscience Department, Research and Development Institute “Life Activities Advancement Institute”, 11000 Belgrade, Serbia; 2Department of Speech, Language and Hearing Sciences, Institute for Experimental Phonetics and Speech Pathology, 11000 Belgrade, Serbia; 3Department of Physiology, Faculty of Medical Sciences, University of Kragujevac, 34000 Kragujevac, Serbia; 4Department of Human Pathology, I.M. Sechenov First Moscow State Medical University, 119991 Moscow, Russia; 5Department of Gynecology and Obstetrics, Faculty of Medicine, University of Montenegro, 81000 Podgorica, Montenegro; 6Clinic of Gynecology and Obstetrics, Clinical Center of Montenegro, 81000 Podgorica, Montenegro

**Keywords:** perinatal mental health, maternal distress, anxiety, stress, depression, developmental outcomes, child development, COVID-19 pandemic

## Abstract

Maternal mental health may be considered a determining factor influencing fetal and child development. An essential factor with potentially negative consequences for a child’s psychophysiological development is the presence of maternal distress during pregnancy and the postpartum period. The review is organized and presented to explore and describe the effects of anxiety, stress, and depression in pregnancy and the postpartum period on adverse child developmental outcomes. The neurobiology of maternal distress and the transmission mechanisms at the molecular level to the fetus and child are noted. In addition, the paper discusses the findings of longitudinal studies in which early child development is monitored concerning the presence of maternal distress in pregnancy and the postpartum period. This topic gained importance in the COVID-19 pandemic context, during which a higher frequency of maternal psychological disorders was observed. The need for further interdisciplinary research on the relationship between maternal mental health and fetal/child development was highlighted, especially on the biological mechanisms underlying the transmission of maternal distress to the (unborn) child, to achieve positive developmental outcomes and improve maternal and child well-being.

## 1. Introduction

The formation of the child’s basic neurological and psychological capacities is influenced by neurodevelopmental, biological, and psychosocial factors [1]. The first thousand days of a child’s life play an essential role in a child’s overall development and later mental health [2]. During prenatal, but especially perinatal and infant development brain adapts in response to a wide range of early experiences, which underlie the creation of developmental pathways, i.e., the development of language, cognitive skills, and socio-emotional competencies [3]. These periods are of significant vulnerability and may be influenced by internal and external risk factors that affect fetal and infant development [4]. The fetal programming hypothesis [5,6,7] explains development modeling and points to the impact of certain events occurring during critical points of pregnancy on permanent effects on the fetus and the infant long after birth. The Developmental Origins of Health and Disease (DOHaD) paradigm and the Developmental Origins of Behavior, Health and Disease paradigm (DOBHaD) point to the interrelation between genes and the environments during the prenatal and early postnatal period [8]. These hypotheses emphasize the interdependence of developmental genetic or environmental influences and suggest that human health and development originate in early life [9]. These paradigms also explain how exposure to stress during specific development time can lead to functional changes in tissues, establishing a predisposition for the appearance of the disease in later life [10]. Namely, the DOHaD hypothesis explains (some) adult diseases as a result of in utero programming in conditions of maternal distress (malnutrition, environmental deprivation). Research shows that fetuses whose mothers were malnourished in the second trimester of pregnancy in adulthood show higher tendencies to renal [11] and pulmonary disease [12]. Infants exposed to mothers’ caloric restriction had a higher incidence of mood disorders [13], cardiovascular disease [14,15], and type II diabetes [11] in adulthood. Some studies pointed out that type 2 diabetes mellitus [16], cancer [17], hypertension [18,19,20], and neurodegenerative disorders [21] are associated with modified DNA methylation mechanisms. Few studies show that maternal protein deprivation results in an increase in epinephrine and norepinephrine that can make lasting changes in infants’ circulation, joined with blood glucose dysregulation, hypertension, obesity, small fetal growth, and altered DNA methylation [19,20,22,23,24]. The studies mentioned above pointed to the significant role of epigenetic mechanisms in the DOHaD paradigm.

The current integrative approach is based on the complex interplay between specific models of developmental stages and possible risks, among which maternal mental illness is considered the developmental risk during early child development [1]. Maternal stress during pregnancy needs to be considered as an important risk factor that can affect fetal and child development and behavior, which has been in research focus for many years [25,26,27,28,29,30]. It can be defined as an experience of general stress, anxiety, depressive symptoms, and adverse life situations [30,31]. A significant number of women experience psychological distress during pregnancy and postpartum. More precisely, the prevalence of maternal anxiety ranges from 6.8% to 59.5% [32], while 10–20% of pregnant women experience psychosocial stress and depressive disorders [33,34]. Accordingly, it could be pointed out that stress, anxiety, and depression are the most common mental health problems during pregnancy [35].

This issue is fundamental during the COVID-19 pandemic, which significantly affects maternal mental health [36]. Recent studies looking at this phenomenon during the COVID-19 pandemic have shown elevated anxiety, depression, and stress levels in pregnant and postpartum women and mothers in the postpartum period [37,38,39,40,41]. Considering the adverse effects of maternal distress on child development, the current increase in mental health problems during the COVID-19 pandemic requires monitoring and support for pregnant and postpartum women, as well as support and stimulation of child development.

Generally, the current review aims to direct attention to new findings on the impact of maternal stress, anxiety, and depression on a child’s development, to promote maternal and child well-being, which may be especially significant due to numerous adverse effects of the COVID-19 pandemic. It is organized and presented to explore and describe the effects of maternal distress in pregnancy and the postpartum period on adverse child developmental outcomes. Hence, it briefly summarizes the results of:The latest research on the neurobiology of maternal anxiety, stress, and depression and the transmission mechanisms at the molecular level to the fetus and child;The longitudinal studies in which early child development is monitored to the presence of maternal distress during pregnancy and the postpartum period.

## 2. Factors and Biological Mechanisms Underlying the Transmission of Maternal Distress to the (Unborn) Child

Fetal development could be considered a sensitive period wherein potential changes in fetal neurobiology may be caused by signals originating from maternal stress experiences and may have a long-lasting effect on the child [42] and developmental programming [43]. Authors Yehuda and Lehrner reviewed the research evidence about parental distress and its possible impact on offspring before birth [44]. The effect of maternal distress on the fetus depends on certain factors which may have a moderating role in the transmission of maternal distress to the (unborn) child. The exposition time [45], period of gestation/pregnancy [46], fetal sex [47], and duration/frequency of maternal distress [48] are factors that may moderate the transmission of maternal stress on the fetus. Some studies pointed out that maternal stress early in gestation may lead to changes in cognitive, behavior, and psychomotor development [49], while other studies pointed to the third trimester of the pregnancy as a specifically vulnerable period during which maternal distress has a more substantial effect on the offspring [47,50,51]. Additionally, the influence of prenatal exposure to maternal distress may be moderated by offspring sex [47,50]. Animal studies demonstrated increased male Hypothalamic–pituitary–adrenal (HPA) stress reactivity, but not female, caused by exposure to chronic stress in utero [52]. On the other hand, human studies demonstrated that female offspring exposed to prenatal maternal stress had higher HPA axis reactivity, with differences in placental expression of *11β-HSD2* (11β-hydroxysteroid dehydrogenase type 2) enzymes, while prenatal stress was associated with alternations in diurnal cortisol secretion in males that were not apparent in females [53]. Recent studies tried to define how prenatal factors could affect gene-specific epigenetic changes in offspring in a sex-specific manner, such as in *IGF2/H19* (insulin-like growth factor 2 gene/H19 gene) [54], *HSD11B2* (Hydroxysteroid 11-Beta Dehydrogenase 2 gene) [55,56], and exon *1F* of *NR3C1* (nuclear receptor subfamily 3 group C member 1 gene) [57,58]. Cao-Lei et al. [47] found a significant interaction effect between maternal anxiety in the third trimester and offspring sex on the methylation level of the CpGs of *IGF2/H19 ICR* and *LINE1* motif2 (the long interspersed nucleotide elements 1), but not of the CpGs of *NR3C1*. This study found a negative association between maternal anxiety and methylation levels in boys and a positive association in girls for *IGF2/H19 ICR*. Furthermore, maternal anxiety in the second trimester also interacts with sex on the methylation level of *LINE 1* motif 2. A previous study revealed a trend-level association between maternal depression and increased *NR3C1* methylation for female infants [50]. Taken together, the sex of the child may be essential for epigenetic processes in early childhood since several effects of the sex of the child on changes in DNA methylation patterns were found as a response to maternal anxiety during pregnancy [47].

Although most studies point to the association between maternal distress and offspring development, further clarification on the different contributions of maternal exposure is needed, including the nature of the exposure, the timing of exposure in pregnancy, the sex of the fetus, the nature of maternal symptoms, nutrition, exposure to toxins, delivery factors, medication effects, socio-demographic variables, and other potential mediators [44].

### 2.1. Mechanisms Involved in Stress Responses

Stress can significantly affect various physiological systems, including the neuroendocrine system, autonomic nervous system (ANS), and immune system [59]. There most often described complex systems that activate neuroendocrine processes as a response to stressors exposure are the hypothalamic-pituitary-adrenal (HPA) axis, the sympathetic-adrenal-medullary (SAM) axis, and the hypothalamic-pituitary-thyroid (HPT) axis.

The HPA axis and the SMA axis are complex systems of neuroendocrine pathways that respond to specific negative feedback loops involving the hypothalamus, anterior pituitary gland, and adrenal gland [60]. The HPA axis presents the major component of the homeostatic response that enables physiological adaptation to the stressor [61]. The activation of both systems is initiated with the release of corticotropin-releasing hormone (CRH) from the hypothalamus, which stimulates the release of adrenocorticotropic hormone (ACTH) from the pituitary gland resulting in the release of glucocorticoid (cortisol) from the adrenal cortex. Cortisol switches off the stress response via a negative feedback mechanism, suppressing the release of ACTH at the pituitary level and CRH at the level of the hypothalamus [62]. The blood concentration of CRH, ACTH, and cortisol increases gradually during the pregnancy, with a rapid increase a few weeks before the parturition [29]. Cortisol level reaches peak concentration in the third trimester of pregnancy [63]. Apart from the importance of the HPA axis during pregnancy, the fetal HPA axis contributes to the constitution of fetal autonomy, having a fundamental role in fetal readiness for birth and survival after birth, while in several species, it determines the timing of birth [64].

At the level of the SAM system, ACTH stimulates the adrenal medulla to release catecholamines (epinephrine and norepinephrine), which are linked to several physiological changes providing an immediate energy source [65]. HPT axes are also involved in stress responses, whereas its final effectors, the Thyroid Hormones (THs), mediate several fundamental processes involved in the neurodevelopment of the offspring [66].

In pregnancy, HPT dysfunction can influence the HPA axis with a direct impact on ACTH and glucocorticoids (GCs) production. While low-circulated THs may decrease cortisol and impair *11β-HSD2* activity, increased THs induce hypersensitivity of cortisol to ACTH [67]. The concentration of GCs and THs must be within the normal range in fetal circulation as they synergistically affect fetal brain maturation and neurodevelopment [66].

Generally, the stress system mainly includes the HPA axis and the ANS [30]. Concerning the sympathetic nervous system (SNS), the primary mediators of the stress system are arginine-vasopressin (AVP), alpha-melanocyte-stimulating hormone, beta-endorphin, and the catecholamines, norepinephrine (NE), and epinephrine (reviewed in Anifantaki et al., 2021) [66].

Several underlying mechanisms of maternal distress transmission to the (unborn) child may operate simultaneously and may amplify each other effects [29]. In the following chapters, the possible mechanisms underlying maternal psychological distress during pregnancy and in the postpartum period will be described.

### 2.2. Biological Mechanisms and Factors Mediating Maternal Distress in the Prenatal Period

Negative psychological experiences during the pregnancy can trigger the expecting mother and affect her biological mechanisms, including over-activation of the HPA axis in response to stress. This leads to the activation of the mechanisms that cause biological changes in the fetal HPA axis, placental function, fetal brain, and the level of some other functions or processes that may underlie fetal programming.

#### 2.2.1. HPA Axis Dysfunction

Both animal and human studies suggest that the HPA axis plays an essential role in mediating the effects of maternal stress on the fetal brain, consequently affecting the brain and behavior of the offspring [68]. In human pregnancy, the maternal HPA axis becomes less responsive to stress as pregnancy progresses due to the human placental production of CRH, which causes an increase in maternal cortisol [69]. Within the HPA axis, CRH plays a central role in the physiologic response to stress [29]. CRH is responsible for preparing the environment for childbirth during pregnancy [70] and is produced and secreted from the paraventricular nucleus of the hypothalamus. Even though it is crucial for fetal growth [71], under particularly stressful conditions, maternal cortisol concentrations can reach abnormally high levels and consequently reach the fetus in high concentrations, which may potentially alter fetal development and growth [72]. Namely, during exposure to stress, altered activation of the HPA axis results in the elevated release of maternal glucocorticoids (primarily cortisol) that enter into fetal circulation above the normal range, thus influencing a range of biological processes and hormonal changes that the placental mechanism includes. Such processes impact the fetal HPA axis functioning, which affects the developing fetus [73], and leads to abnormalities in cell structure formations, neurotransmission, a function of the fetal central nervous system [74], and may have a long-lasting influence on child development [75]. Finally, early life exposure to excess fetal glucocorticoid (GC) hormones caused by maternal distress can alter normal neuropeptide synthesis and lead to a disruption in the development of the fetal HPA axis, leading to abnormalities in neuroendocrine, behavioral, autonomic, and metabolic functions in adulthood [61].

#### 2.2.2. Placental Mechanism

The placenta is an effective physiological barrier between the maternal and fetal hormonal environments in humans, and it controls fetal exposure to placental and maternal hormones (glucocorticoids, including cortisol) and environmental factors [71]. In pregnancy, from about 8–10 weeks gestation, CRH is also produced by the placenta and has the same biological activity as hypothalamic CRH [29]. Placental CRH (pCRH) is determined to have a major role in mediating the effect of maternal stress on the (unborn) child [76], with secretion to both maternal and fetal compartments. Overproduction of fetal cortisol may arise from maternal cortisol in a fetal compartment and/or from pCHR secretion [29]. Furthermore, maternal stress or anxiety can induce increased transplacental transfer of maternal cortisol to the fetal compartment without an increase in maternal levels [69].

The placenta contains protective enzymes such as monoamine oxidase A, peptides, and 11β-hydroxysteroid dehydrogenase type 2 (*11β-HSD2*). The *11β-HSD2* is an enzyme that converts cortisol to inactive products such as cortisone and may act as a barrier in transferring elevated levels of GCs from maternal to fetal circulation [77]. Namely, one of the primary roles of the placenta is controlling fetal exposure to maternal cortisol via enzyme *11β-HSD2* [66]. Although there are two forms of *11β-HSD* (type 1, *11β-HSD1*, and type 2, *11β-HSD2*), having significant roles in the bioactivity of glucocorticoids, the predominant form expressed in human placenta is *11β-HSD2* maintaining a concentration gradient between the cortisol levels in the mother’s and the child’s compartments [78]. During pregnancy, *11β-HSD2* expression in the placenta regulates the level of active GCs through the oxidation of maternal corticosteroids into inactive 11-keto derivatives, reducing fetal exposure to active GCs [61]. More precisely, *11β-HSD2*, as NAD+− dependent, unidirectionally catalyzes active corticosterone and cortisol conversion to inert 11− dehydrocorticosterone in rodents and cortisone in humans [78]. Thus, as pregnancy progresses, placental *11β-HSD2* constitutes a specific barrier described as the placental glucocorticoid barrier and restrains the transfer of maternal cortisol (whose concentrations in a pregnant woman are several times higher than those observed in the developing fetus) to the fetus, also initiating the development of the fetal HPA axis [78,79]. When a mother is stressed in a way that increases her cortisol level, inactivity of the placental enzyme *11β-HSD2* may leave the fetus vulnerable and less protected from the mother’s circulating hormones, consequently reflecting the hormonal milieu of the fetus [7,71]. Studies in humans demonstrated that maternal distress downregulates the expression and activity of *11β-HSD2*, resulting in an alternation in the filtering capacity of the placenta [80] and abnormally elevated levels of CGs in a fetal environment [69]. Recent studies suggest that *NR3C1* (nuclear receptor subfamily 3 group C member 1), which is highly expressed in the placenta due to maternal distress, may be an upstream regulator of placental *11β-HSD2* gene expression, increasing the placental sensitivity to GCs [81].

Maternal distress transmission on the level of placental and breastfeeding mechanisms is shown in Figure 1.

Distress can have an impact on the (unborn) child on two levels:Prenatally, the fetus could be affected while still in the uterus via the placenta and dysregulated maternal HPA axis;Postnatally, the newborn infant could be affected via breastfeeding and changed milk composition (disrupted concentrations of hormones, immune cells, and other components).

#### 2.2.3. Catecholamine, Uteroplacental, and Fetal Hemodynamics

Maternal distress may also cause a reduction in the fetoplacental blood supply which is a critical determinant of placental function and fetal growth and another essential part of a placental mechanism [30]. A recent animal study demonstrated that chronic prenatal psychosocial stress during the first and second trimesters could increase fetal catecholamine concentration [45]. The release of stress hormones, such as corticosteroids and catecholamines, strongly affects the tone of peripheral blood vessels [29]. These hormones, which are products of the SAM axis, stimulate α and β neuroreceptors in the placenta, resulting in vasoconstriction of the uteroplacental vessels. Norepinephrine stimulates α-adrenoceptors leading to increased blood pressure, constriction of blood vessels, and decreased uterine blood supply. On the other hand, norepinephrine increases the production of pCRH and causes a series of endocrine interactions that may affect pregnancy and fetal development [60].

#### 2.2.4. Immune System and Inflammation as Maternal Mediators of Stress

There is evidence of mediating role of the immune system and inflammation in the transmission of maternal stress [68,69]. Pro-inflammatory cytokines and their altered patterns during pregnancy have been associated with maternal psychosocial stress, depression, or anxiety [82,83]. Elevated maternal stress in early and late pregnancy has been associated with elevated production of the pro-inflammatory cytokines interleukin-6 (IL-6), interleukin-1β (IL-1β), and tumor necrosis factor (TNF-α), suggesting that stress affects immune system cells [69,84], and may be associated with increased risk of fetal outcomes [85]. Studies examining cytokines’ role in mother–fetus interaction demonstrated that pro-inflammatory cytokines such as TNFα, IL-1β, and IL-6 do not cross the placenta during normal pregnancy [86,87]. On the other hand, increased TNFα, IL-1β, and IL-6 in the amniotic fluid expresses an inflammatory process, which may lead to an increased risk of premature delivery, cerebral palsy, and bronchopulmonary dysplasia in offspring [86].

Evidence on this topic is limited and inconsistent, derived mainly from animal models [88].

#### 2.2.5. Serotonin and Tryptophan

Data about the neurotransmitter serotonin (or 5-hydroxytryptamine, 5-HT) and tryptophan in maternal–fetal stress transmission is limited, especially in humans. Serotonin is synthesized from tryptophan and has a crucial role in fetal brain development [89]. Studies in animals pointed to the potential role of serotonin and tryptophan in fetal programming [90]. Molecular alterations observed in animal models exposed to prenatal maternal stress include: turnover of serotonin (5HIAA/5-HT ratio, (5-hydroxyindoleacetic acid/serotonin)) in the hippocampus, increased TPH (tryptophan hydroxylase) expression, and serotonin level in the dorsal raphe nuclei, increase the 5-HT1A receptor (serotonin 1A receptor) in the hippocampus, and the decrease in SERT (serotonin transporter or 5HTT) levels in the offspring who may have a manifestation of depression in later life (reviewed in St-Pierre et al., 2016) [89]. Maternal depression during pregnancy may be associated with the downregulation of enzyme monoamine oxidase A (MAO A), thus affecting the transplacental serotonin passage from the mother to the fetus [91]. Thus, the potential programming role of serotonin may be defined by the modification of the serotonin systems (maternal, placental, and fetal serotonin systems) caused by maternal distress during a sensitive period of early development, which may affect fetal development and increase the risks of anxiety, depression, and autism later in life [30,89,92].

#### 2.2.6. Oxidative Stress: Interaction between Maternal and Fetal Oxidative Systems

Fetal programming may be influenced by maternal oxidative stress and reactive oxygen species (ROS), which in excessive concentrations, can induce severe cell damage by oxidation of lipids, proteins, and DNA (deoxyribonucleic acid) [88]. The modification of the fetal epigenome is caused by the intracellular ROS, which decreases the activity of the nuclear DNA epigenetic control mechanisms inducing global DNA-hypo-methylation [93] and interfering with the control of histone methylation [94].

#### 2.2.7. Neuroactive Steroids

The primary neuroactive steroids proposed to play a role in fetal programming are allopregnanolone and 5α,3α-tetrahydrodeoxycorticosterone (THDOC). Both are potent modulators of GABAergic inhibitory and neuronal functions and modulate HPA axis function. It was demonstrated that stress could reduce circulating levels of allopregnanolone at birth, resulting in reduced production in the offspring’s brain. The reductions in neuroactive steroids may be linked with cognitive impairments and anxiety-evoking effects [88].

#### 2.2.8. Maternal Microbiota as a Potential Stress-Transfer Mechanism

Maternal microbiota and its vertical transmission to the fetus or newborn is rather hypothetical than demonstrated stress-transfer mechanisms [88]. This mechanism refers to the programming of the fetal gut–brain axis caused by impaired maternal microbiota, which is linked to maternal stress. The bi-directional communication system between the gastrointestinal tract and the brain (so-called gut–brain axis) includes neuronal, immune, and endocrine pathways (involving cytokines, neurotransmitters, and other neuromodulators) and potentially is associated with neurodevelopmental and psychiatric diseases [30,45]. The stress-induced modulation of the ‘gut-brain’ axis changes the enteric synthesis of neuroinflammatory cytokines, neuromodulators, and neurotransmitters involved in the newborns’ neurodevelopment, which may lead to a greater susceptibility to neuropsychiatric disease in later life [95].

#### 2.2.9. Autonomic Nervous System

Apart from the HPA axis, the autonomic nervous system is the second major stress response system. The ANS is divided into two main branches: the sympathetic nervous system (SNS) and the parasympathetic nervous system (PNS), which regulate heart rate (HR). Heart rate and heart rate variability (HRV) have been associated with both higher and reduced baseline fetal HR in pregnancies burdened with maternal prenatal stress, anxiety, and depression (reviewed in Vehmeijer et al., 2019) [30]. On the other hand, the involvement of ANS activity in the maternal–fetal stress transfer mechanism is demonstrated in increased catecholamines which induce vasoconstriction of the uteroplacental vessels and consequently lead to the reduction of uteroplacental blood flow, which is previously described. There is no conclusive evidence of a clear association between maternal and fetal autonomic functioning in the presence of maternal stress during pregnancy and their potential links with fetal/child health outcomes [30,51].

#### 2.2.10. Gene–Environment Interactions, Epigenetics, and Prenatal Stressors

The development of the fetus’s brain regions and cognitive-emotional functions is assumed to depend on gene–environment interactions [69]. Some children are affected by the various forms of prenatal stress to develop later child psychopathology, while some are not. The “genetic makeup” of the child related to environmental factors is, at least partially, the reason for this distribution [69]. This is under the differential susceptibility framework of gene–environment examinations, which suggests that an individual’s biological context moderates sensitivity to both positive and negative environmental influences; and views traditionally labeled “vulnerable” individuals as “plastic/malleable” individuals [1].

Gene–environment interactions may cause epigenetic changes in gene expression patterns and thus affect or create a specific phenotype of human psychophysiological abilities or disease. Namely, the genome, epigenome, and environment act simultaneously and thus create certain phenotypic traits in humans [96]. The suggested ‘Fetal programming’ mechanism, according to the DOHaD hypothesis, points to a relationship between adverse environmental effects and genetic and epigenetic modifications during critical prenatal and later human health or disease [47]. The term “epigenetic” refers to alterations in gene function in the absence of changes in the DNA sequence. In contrast, the term “epigenom“ refers to chemical modifications to DNA, chromatin, and histone structure, nucleosome positioning, and functions of non-coding ribonucleic acid (RNA) [97]. Furthermore, the three primary molecular epigenetic mechanisms comprise DNA methylation, histone modifications, and non-coding RNAs. DNA methylation refers to the covalent modification of cytosine with a methyl group, histone modification includes acetylation, methylation, phosphorylation, and ubiquitination, while microRNA refers to small non-coding RNAs with post-transcriptionally regulated gene expression [98].

##### DNA Methylation

DNA methylation (DNAm) is a biological process involving a post-replication enzymatic modification of DNA, which refers to binding the methyl group to carbon in position five of the cytosine ring. It can change the activity of a DNA segment without changing the sequence. Under normal conditions, DNA methylation is essential for normal development. It is associated with several critical processes involved in the transcriptional silencing of genes, regulation of expression of imprinted genes, some tumor suppressor genes in cancer, and silencing of genes located on the inactive X chromosome [99]. On the other hand, DNA methylation may be modified by prenatal maternal and fetal stressors, thus affecting downstream gene expression, neuroendocrine functioning, and behavioral development [98].

Several genes are involved in response to stress via the HPA axis, autonomic nervous system hyperactivity, and cortical and subcortical processes of neuroplasticity and neurodegeneration based on changes in the scope of their expression. These genes have been targeted based on the hypothesis that methylation alterations mediate stress’s biological effects. These genes include: the corticotrophin-releasing hormone gene (*CRH* gene), arginine vasopressin gene (*AVP* gene), glucocorticoid receptor gene nuclear receptor subfamily 3 group C member 1 (*NR3C1*), FK506 binding protein 5 gene (*FKBP5*), Solute carrier family 6 member 4 gene (*SLC6A4*) which is a serotonin transporter, also known as the *5-HTT* gene, and brain-derived neurotrophic factors [96,100].

Furthermore, infant neural functions may be directed by the influence of the prenatal emotional disturbances of the mother via DNA methylation of the 11β-hydroxysteroid dehydrogenase type 2 (*11β-HSD-2*) and glucocorticoid receptor nuclear receptor subfamily 3 group C member 1 (*NR3C1*), which are two placental genes that have been implicated in perturbations of the HPA axis [101].

The *NR3C1* gene and the region of exon 1F have been extensively studied regarding disturbed emotional states (depression and anxiety) during pregnancy [47,101,102,103] and associated with DNA methylation changes (higher methylation levels) of placental *NR3C1* exon *1F*. Apart from the *NR3C1*, in the study of Martinez et al. [70], a significant increment in the gene expression of the hydroxysteroid dehydrogenase (*HSD11B2*) encoded by the enzyme 11ß-HSD2 was demonstrated, which consequently may negatively affect the neurodevelopmental outcomes of the newborn. The placental 11β-HSD2 enzyme converts active cortisol into inactive cortisone, thus regulating the degree to which cortisol can pass through the placenta and protecting the fetus from excessive glucocorticoid exposure [80]. Increased placental DNA methylation led to decreased expression of 11β-HSD2 in infants with the lowest birth weight [101,104]. Modified gene expressions can consequently be associated with adverse effects on poor newborn neurodevelopmental outcomes, such as lower birth weight and reduced newborn movement quality [70,105].

Increased placental FK506 Binding Protein 51 (*FKBP51*) methylation is associated with reduced fetal coupling and potentially predicted neurobehavioral development of the newborn due to higher perceived stress in the mother [104].

Among the other relevant genes enrolled in prenatal stress is Solute carrier family 6 member 4 (*SLC6A4*). It is a widely studied gene that encodes serotonin transporter [98]. Maternal depressive symptoms during pregnancy can be associated with decreased gene methylation status in children [106].

There are limited findings about oxytocin receptor gene (*OXTR*) methylation caused by maternal distress. Both a decrease and increase in infant *OXTR* methylation were associated with maternal distress (reviewed in Sosnowski et al., 2018) [107].

Furthermore, the brain-derived neurotrophic factor (*BDNF*) plays a role in neural cell growth, maturation, and maintenance. The opinions are opposite, but some studies detected a significant association between prenatal depressive symptoms and decreased BDNF promoter IV DNA methylation in buccal DNA in infants [98].

DNA methylation of specific imprinted genes may also be associated with maternal mental health during pregnancy. The insulin-like growth factor 2 gene (*IGF2* gene) plays an essential role in fetal growth and development, while the *H19* gene is located in an imprinted control region (*ICR*) of chromosome 11 near the *IGF2* gene and is expressed from the maternally-inherited chromosome. There are inconsistent findings indicating both increased and decreased DNA methylation of *IGF2/H19 ICR* in the placental and cord blood of children exposed to higher levels of maternal anxiety [54,108], additionally influenced by the sex of the child [47]. There are other imprinted genes such as *MEG3* (Maternally Expressed gene 3), *PLAGL1* (PLAG1-Like Zinc Finger 1), *PEG3* (Paternally Expressed 3) associated with DNA methylation in offspring who experienced prenatal maternal distress (reviewed in Cao-Lei et al., 2020; reviewed in Ryan et al., 2017) [73,98].

The long interspersed nucleotide elements 1 (*LINE1*) as an indicator of global DNA methylation status was also examined concerning DNA methylation in children whose mothers experienced high levels of distress during pregnancy. Cao-Lei et al. (2021) [47] found higher methylation in boys and lower in girls for *LINE1* motif2 exposed to prenatal maternal distress.

##### Histone Modification

Histone modification is an epigenetic mechanism for regulating gene expression, which may be induced by the response to DNA methylation or by intracellular signaling pathways independent of DNA methylation [97]. Several histone modifications include acetylation, phosphorylation, methylation, deimination, ADP-ribosylation, ubiquitylation, sumoylation, and proline isomerization [109]. This mechanism affects the chromatin structure, which plays an essential role in DNA packaging, gene transcription, and cross-talk between DNA methylation and chromatin configuration [97], and may lead to the activation or suppression of gene expression. In animal studies, histone modifications have been reported to be crucial targets for interacting with the early stress response system and manifesting later depressive and long-term adverse effects such as preclinical and clinical development of depression, anxiety, depression-like behaviors, and development of psychiatric diseases, such as major depressive disorder in adulthood [110,111].

##### Changes in Non-Coding RNAs

Non-coding RNAs (ncRNAs) are untranslated RNA molecules that regulate gene expressions. These are small nuclear RNAs (snRNAs), small nucleolar RNAs (snoRNAs), ribosomal RNAs (rRNAs), transfer RNAs (tRNAs), circular RNAs (cRNAs), and piwi-interacting RNAs (piRNAs) [112]. In a recent review, it was demonstrated that ncRNAs might impact the epigenome in the context of different environmental risk factors, including mental stress, and contribute to the pathophysiology state [113].

Finally, despite large evidence to suggest that epigenome corresponds with altered gene expression in children exposed to maternal distress during pregnancy, there is a lack of reproducible findings and potential limitations, which imposed the need for further research in this research field.

#### 2.2.11. Neurodevelopmental Mechanisms

The transmission of maternal stress hormones, especially glucocorticoids, across the placenta to the fetus may play a critical role in fetal neurodevelopment [1]. Evidence indicated that overexposure to maternal glucocorticoids caused by downregulation of 11β-HSD2 mRNA expression and activity is a significant predictor of spontaneous preterm birth and low birth weight [114]. Higher levels of maternal cortisol tend to cause alterations in the neural activity of the babies, and it refers to increased low-frequency brain activity (i.e., higher relative theta power) and decreased high-frequency brain activity (i.e., less relative alpha and higher-gamma power and marginally less beta power). This study indicates that a mother’s chronic psychological stress negatively affects the newborn’s developmental patterns of brain activity [115]. Maternal anxiety during pregnancy may be related to specific changes in brain morphology depending on exposition time. It was demonstrated that maternal anxiety at 19 weeks gestation has been associated with grey matter volume reductions in the prefrontal cortex, the premotor cortex, the medial temporal lobe, the lateral temporal cortex, the postcentral gyrus, and the cerebellum extending to the middle occipital gyrus and the fusiform gyrus [116]. A recent study examining the association of prenatal maternal psychological distress with fetal brain growth, metabolism, and cortical maturation found impaired fetal brain biochemistry and hippocampal growth, and accelerated cortical folding [117].

On the other hand, maternal anxiety during pregnancy has been linked to problems in infant temperament, behavior, and cognitive development; emotional and behavioral problems in children and adolescents; and structural brain [1]. In the review article of Van den Bergh et al. [7], the studies that have observed fetal behavior of highly anxious or stressed mothers were presented; the results indicated more bodily activity and spontaneous motor activity in the observed fetuses. Furthermore, these infants exhibited more frequent behaviors such as crying, irritability, irregularity of biological functions, gripes, and difficult temperament, postnatally. Male children from these studies exhibited increased activity, attention deficit, and aggressiveness at nine years.

### 2.3. Biological Mechanisms Underlying the Maternal Distress Transmission to the Child in the Postpartum Period

Biological mechanisms underlying maternal distress transmission to the child in the postpartum period are described through breastfeeding, mother-child bonding, and caregiving mechanism.

Breastfeeding appears to be very beneficial for the mother and her child. However, postpartum depression might have adverse effects on maternal self-esteem and cognition. Inadequate bonding between women with depressive symptoms and their newborns increases the risk of breastfeeding difficulties [118].

Breast milk is the best food for infants, containing all the nutrients they need for normal development and health. Breast milk composition presents a mediatory mechanism connecting perinatal psychopathology with a child’s brain development. Maternal perinatal psychopathologies, such as depression, anxiety, and stress, determine strong biological alterations in the affected women, which can change breast milk’s regular composition (Figure 1). Given that breast milk has an essential role, an alteration in its composition can severely affect child neurodevelopment, impairing fundamental functions related to cognition, behavior, and attention [119].

#### 2.3.1. Effects of Maternal Distress on Milk Composition

In the study by Semir Demirgoren et al. [120], it has been shown that mothers with postnatal depression and state anxiety had higher than expected breast milk Na concentrations and a high NA/K ratio (sodium levels and sodium/potassium ratio in breast milk). The anxiety symptoms that appear postnatally in a specific moment (state anxiety) can affect more significantly the composition of breast milk compared to the woman’s personality and proneness to anxiety (trait anxiety) [119]. This study confirmed that increased breast milk Na and Na/K ratio is associated with mothers’ depressive and anxious symptoms in the postpartum period, with possibly severe consequences in infants [120].

However, Kim et al. [121] have recently found that no endocrine-disrupting chemicals in milk were significantly associated with developing postpartum depression. The authors have noted that concentrations of mono-2-Ethylhexyl phthalate and ethylparaben in breast milk were positively associated with the risk of postpartum depression in the group of Korean mothers.

Nevertheless, the information about toxic metal concentrations in breast milk is limited. The association between metal breastmilk concentration and maternal distress in the postpartum period should be further investigated.

Shariat et al. [122] have found a significant positive association between milk transforming growth beta factor-2 (TGFβ2) concentration and both postpartum depression and anxiety.

Secretory IgA (sIgA), a vital component in the first-line defense of the immature immune system and microbiota development in early life, was significantly decreased in infants born to mothers with pre and postnatal depressive symptoms. This study suggested that lower sIgA concentrations in these infants predispose them to a higher risk for allergic disease [123].

Following these results, Kawano and Emori [124] found a negative association between maternal postpartum psychological state and breast milk sIgA levels, concluding that maternal psychological state may affect the immune components of breast milk.

Furthermoref, another study showed that postpartum stress could be associated with reduced sIgA concentration, suggesting that women may have reduced breast milk immunological benefits, thus offering less immunological protection to their infants [125].

However, the authors Aparicio et al., 2020 found that maternal psychosocial distress was positively related to higher milk cortisol concentrations at week two post-delivery but generally concluded that there is no evidence for an association between natural variations in maternal distress and immune factor concentrations in milk [126].

Based on the evidence from their recent study, the authors suggested that the mother’s psychological well-being may be related to the immunological components of her milk, according to the fact that increased anxiety negatively affects milk’s immune properties. Concerning positive mental health factors, maternal social support was found to be correlated with milk IgG [127]. Pregnancy and lactation may exacerbate low Docosahexaenoic acid (DHA) status because maternal DHA stores are mobilized to support the rapid development of the fetal and infant brain [128].

Rather than transient depressive symptoms during the pregnancy, low breast milk DHA may reflect chronic depression more closely, given that breast milk DHA levels largely reflect fatty acid stores laid down over many years. The association between depressive symptoms early in pregnancy and reduced DHA breast milk levels appeared significant in the study of Keim et al. [129].

Ziomkiewicz et al. [130] found a significant and negative effect of maternal stress during the postpartum period on the composition of breast milk. Perinatal psychosocial stress negatively affected energy density, fat, and medium-chain and long-chain saturated fatty acids in milk. Higher cortisol level secreted was positively associated with the content of long-chain mono- and polyunsaturated fatty acids and lower lactose content in milk.

The predictor of severe depressive symptoms in postpartum women was higher thyroxine levels, while the predictor of depression appearing 6–10 weeks postpartum was higher progesterone and lower prolactin levels. Women who had previous episodes of depression had decreased prolactin and increased thyroid-stimulating hormone levels than those who had experienced depressive symptoms before [131].

In a prospective cohort study, Stuebe et al. [132] found that higher depression and anxiety symptoms were correlated with lower levels of oxytocin in response to breastfeeding at eight weeks postpartum. These results suggest that maternal psychological symptoms are associated with neuroendocrine changes in lactation.

According to a few studies, there are indices that maternal distress can be associated with changed maternal milk composition; however, this field is not yet explored enough. The mechanism of transferring through breastfeeding is possible, and molecular pathways are the direction for future examinations.

A positive relation was found between maternal psychosocial distress and breast milk cortisol levels in the early postnatal period [126]. A recent study revealed that preterm infants are born deficient in transplacental hormones such as TSH, thyroxine, and albumin because their intrauterine development is interrupted early [133]. Furthermore, another study revealed that stress reactivity was associated with milk components, such as milk energy density, fat, medium-chain, and long-chain saturated fatty acids [127,130,134]. A recent study by Kortesniemi (2021) found that the human milk metabolome is associated with maternal psychological distress and milk cortisol concentration, pointing to stress-induced changes in the microbiome–gut–brain axis and energy metabolism [135].

#### 2.3.2. Mother-Child Bonding and Caregiving Mechanism

Maternal distress has multiple effects on a child’s development, including the neurodevelopmental impact of stress-related hormones and the psychosocial impact of parenting behavior. This focus is central to the development of integrative models for the development of psychopathology [1].

Attachment theory proposes an integrative framework of human development, where development occurs in the context of early relationships, which provide security and comfort [136].

Cooke et al. [137] investigated maternal attachment insecurity and depression symptoms and acted as an intermediary mechanism of mothers’ adverse early experiences contributing indirectly to the intergenerational transmission of risk for children’s behavioral and psychological symptoms. Both maternal attachment avoidance and attachment anxiety in adulthood contributed to these pathways. In contrast, maternal anxiety symptomatology was not found to significantly mediate these pathways [137].

The immature brain of the infant reacts differently to various ways of stimulation or insufficient stimulation, and these effects are related to elevated catecholamines, delays in myelination, and synaptic pruning [1].

Indeed, a growing body of evidence suggests the epigenome is also responsive to social and environmental exposures due to its role in cellular programming, during intra-uterine development and after birth. Exposure to stress early in life may affect the epigenome, but these effects may be silent depending on whether the system is triggered or not throughout life by any factors [138].

The authors studied the impact of maternal care on the epigenetic programming of the NR3C1 gene promoter in the rat hippocampus and found that DNA methylation patterns of the NR3C1 gene can be altered after birth in response to the social environment [138,139].

## 3. Maternal Distress as a Risk Factor for Adverse Fetal and Child Developmental Outcomes

From all the above, it follows that maternal distress is a risk factor that can significantly affect fetal psychophysiological development and is often positively correlated with pre and perinatal risk factors. Recent research shows that elevated levels of maternal stress, anxiety, and depression during pregnancy are associated with the risk of preeclampsia, prematurity, low birth weight, and poorer neonatal outcome [140,141,142].

Furthermore, it is essential to point out that maternal distress may often be associated with specific changes in brain morphology and abnormalities in the functioning of the central nervous system [116,143]. These specific changes in brain function and morphology lead to vulnerable early development of the human individual, which is re-percussed on later development as well. Maternal prenatal distress affects the outcome of the pregnancy and results in early programming of fetal brain functions with permanent changes in neuroendocrine regulation and offspring behavior [68,69]. The impact of maternal distress during pregnancy on fetal outcomes and child cardio-metabolic, respiratory, atopic, and neurodevelopment-related health outcomes is also well documented [30]. Generally observed, the underlying risk factors caused by maternal distress are associated with various aspects of further child development, such as cognition, speech-language development, motor development, behavioral and learning difficulties, socio-emotional development, and neurodevelopmental disorders [9,144,145,146].

### 3.1. Cognition

Cognitive development is a complex mental process based on the ability of the comprehension and process information. The lower mental development of children might be affected by mothers’ prenatal exposure to distress. Several studies found that women exposed to stressful life events such as natural disasters as well as prenatal distress had a significantly increased risk of children with poorer cognitive development at different ages [147,148,149,150]. Trait anxiety in the second trimester and both trait and state anxiety at 32 gestation weeks predicted lower mental developmental scores in children aged 8 and 24 months [147,148]. Elevated maternal cortisol levels in late pregnancy were associated with lower mental development in children aged 3 months [148]. Unlike previous studies, the study of Keim et al. [151] indicates that pregnancy-specific anxiety very poorly correlates with overall mental development at 12 months of age.

Studies dealt with the influence of maternal distress during pregnancy on language development as well as language skills such as reading, spelling, writing, and mathematics skills (numeracy), most often comparing the impact of mothers’ distress on the development of receptive and expressive language in children. Research findings pointed to lower productive language abilities and receptive language in toddlers aged 30 months who were prenatally exposed to maternal distress [149]. On the contrary, a study conducted by Keim et al. [151], in which the association between prenatal and postnatal anxiety, stress, and depressive symptoms on language ability of 12-month-old infants was examined, pointed to better language skills in children whose mothers had higher depression symptoms. The impact of prenatal maternal distress on specific learning abilities (literacy and numeracy) in children aged 10 was examined in a large population follow-up study [152]. This study pointed to the significant difference between genders: girls had lower accomplishments in reading, while no association was found between prenatal distress and spelling, writing, and numeracy, compared to boys whose mothers experienced maternal distress. On the other hand, boys whose mothers had three or more stressful events during pregnancy had higher scores on numeracy and writing test compared to boys from the control group.

### 3.2. Socio-Emotional Development

Socio-emotional development is a complex process that involves the interaction of maturation factors and the child’s environment. Some studies reported an association between maternal distress during pregnancy and postpartum and poorer socio-emotional development [75,153,154]. Further, a small study of mother-infant dyads [143] used diffusion tensor imaging (DTI) pointed out the association between the internalizing domain of behavior and neonatal insula variation in infants whose mothers had prenatal distress.

Numerous studies have described the impact of maternal distress on various aspects of temperament [150,155,156]. Infants between 16–18 months whose mothers experienced more stressful life events during early pregnancy (first trimester) had low regularity, persistence, and attention span [150]. Likewise, pregnancy-specific anxiety was a positive predictor for 6-months-old infant fearfulness and falling reactivity [155], while infants whose mothers were exposed to higher levels of distress during the first trimester of pregnancy (illness/infection conditions, older mothers, subject distress) were found less responsive [157]. Furthermore, at six months of age, pregnancy-related anxiety can be a predictor of mood disturbance [140].

Trait anxiety in late pregnancy predicts a more difficult temperament in infants, while state anxiety and pregnancy-related anxiety had no significant effect on temperament [158].

Few studies tried to determine whether mothers’ distress during pregnancy predicts the value of mother-infant bonding postpartum. During the COVID–19 pandemic, the study examined mother-infant dyads for 3 months, and the results showed that lower levels of maternal anxiety at delivery are correlated with higher mother-infant bonding. On the other hand, in another study, a positive correlation was found between prenatal pregnancy anxiety, postpartum trait anxiety, postpartum depression, and postpartum bonding, but a significant association was found between pregnancy-related fear, postnatal anxiety, and mother-infant bonding [159]. One particular study examined the multimodal processing of face and voice pairs regarding fearful and happy emotions in 9-month-old infants [160]. The authors hypothesized that non-typical experiences, such as mothers’ distress during fetal development, can alter the development of this ability, and the results of the study showed that general anxiety, but not state anxiety, had significant effects on the processing of emotional information.

### 3.3. Fine and Gross Motor Development

Motor development involves the development of two aspects of motor skills: fine and gross motor skills. Studies assessing infants’ motor development at 24–30 months [75] and 36 months of age [161] indicated that high prenatal maternal anxiety was associated with poorer gross motor development related to delays in fine and gross motor development and problem-solving domains.

Furthermore, fine and gross motor development was assessed in children at 2, 6, and 16 months [146]; 10 months [162]; and 12 and 24 months [147]. At 2 months, fine and gross motor skills were positively correlated with maternal peritraumatic distress levels. However, at 6 months, fine and gross motor skills were negatively correlated with maternal distress. No significant association was found between fine gross motor functioning and maternal distress at 16 months of age, while gross motor skills were negatively correlated with maternal distress [146]. Further, at 10 months of age negative correlation was found between maternal trait anxiety and fine motor skills, while no significant correlation was found between maternal anxiety and gross motor skills, visuomotor function, and brain stem function [162]. Moreover, at the ages of 12 and 24 months, the motor development of infants who were exposed to maternal distress during pregnancy was observed within the overall development [147]. This study’s results indicated lower scores on the motor scale in infants exposed to higher maternal distress at the ages of 12 and 24 months.

The longitudinal study assessed the effects of objective and subjective prenatal maternal stress of pregnant women exposed to a natural disaster on the motor development of 5½ years old children and showed a correlation between prenatal maternal stress and bilateral coordination and visual-motor integration [163]. Finally, the study investigated the long-term effects of the number and timing of stressors experienced during pregnancy on motor development at 10-, 14- and 17-years old children [164] and indicated that children whose mothers had more than three distressing events during pregnancy had lower motor competence at all tested ages. This result implicated that number of experienced prenatal stresses may have a cumulative effect on offspring motor development.

### 3.4. Neurodevelopmental Disorders

The most frequent neurodevelopmental disorders, which are characterized by developmental deficits that produce impairments of personal, social, academic, or occupational functioning, are: attention-deficit/hyperactivity disorder (ADHD) and autism spectrum disorder (ASD) [165]. Several studies have been investigating maternal prepartum and postpartum distress as risk factors for ASD and ADHD [166,167,168], pointing to the association between prenatal stress and a higher frequency of diagnoses of ASD or ADHD in children after the age of 3. Gender differences (for ADHD diagnosis) [168] and time of exposure to stress during pregnancy were found to be significant for both ASD (first and third trimester) and ADHD (third trimester) [167]. Some studies focusing on the symptoms of ASD and ADHD in children whose mothers experienced prenatal distress showed that both boys and girls, not gender-related, at the age of 6 years and 6 months scored higher on autism-like traits, especially when mothers experienced more than one stressful event during pregnancy [169] while only boys were likely to be at a higher risk for ADHD symptoms [170]. On the other hand, a study focusing on the diversity of autism symptoms in children already diagnosed with ASD who have been exposed to distress during intrauterine development [171] found that the symptoms of autism are more severe only in those children whose mothers experienced more than one stressful event during pregnancy. The symptoms were significantly increased in the domain of behavior (repetitive and restricted behavior), communication, and language (syntax, semantics, coherence, and stereotyped language). The obtained results may indicate that prenatal stress can alter ASD phenotype.

Reviewed studies implicate the association between prenatal maternal distress and specificity/delays in cognitive, socio-emotional, and motor development in infants and children, but neurobiological mechanisms underlying these developmental delays need to be further examined.

## 4. Conclusions

In addition to the well-known adverse effects of biomedical risks, the mother’s psychological factors can significantly contribute to complications in pregnancy and the unfavorable development of the (unborn) child. Maternal distress must be monitored prenatally and in the postpartum period concerning specific biological transmission mechanisms. This topic gained importance because of the numerous adverse effects of the COVID-19 pandemic, during which a higher frequency of maternal psychological disorders was observed. The need for further interdisciplinary research on the relationship between maternal mental health and fetal/child development was highlighted, especially on the biological mechanisms underlying the transmission of maternal distress to the (unborn) child, to achieve positive developmental outcomes and improve maternal and child well-being.

## Figures and Tables

**Figure 1 ijms-23-13932-f001:**
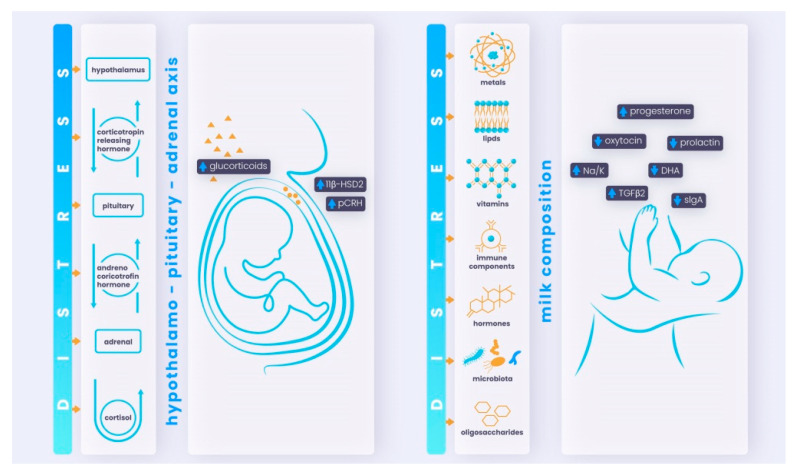
Maternal distress transmission: placental and breastfeeding mechanisms.

## Data Availability

Not applicable.

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
