# Peer review of "Maternal Distress during Pregnancy and the Postpartum Period: Underlying Mechanisms and Child’s Developmental Outcomes—A Narrative Review"

_ijms, 2022, doi:10.3390/ijms232213932_

Round 1
Reviewer 1 Report
In this manuscript, the authors reviewed recent studies to explore and describe the effects of anxiety, stress, and depression in pregnancy and the postpartum period on adverse child developmental outcomes. This is a very well written and interesting review. I recommend acceptance.
Author Response
Thank you for your valuable opinion and recommendation for acceptance.

Reviewer 2 Report
The review is well written and organized, and it is focused on the effect on the risk of maternal stress on the adverse child developmental outcomes. It is an important topic especially nowadays in which COVID-19 pandemic has affected the mental health of the world's human population. The mental health of pregnant women is an aspect that needs attention for the impact that it cause on the development of chidren in early life and an in their later life.
Despite the interesting multi disciplinary approach of this review, I found that the treated biomedical part needs some reviews, and the excess of fragmentation in subheadings does not help in the comprehension of the text.
As just an example, all the part between 2.3.1-2.3.4 subheadings may be summarized as: effects of maternal distress on milk composition. Also the title of the 2.3 part may be reduced as: biological mechanism underlying rhe maternal distress transmission to the chinìld in the postpartum period. The rest of the title may be postponed in the subsequent paragraphs.
Few minor changes
- lines 78-82: perhaps a bulleted list of the content of the work may be useful, as it is quite a long explanation.
- lines 82-85: I would mover this paragraph over the previous lines, at the beginning of this paragraph.
- line 92: an emerging body of literature... and then you cited just one article at the end of the sentence (30). You probably need to put here all the articles that are focused on the topic of parental distress and its effect on offspring before birth.
- chapter2, lines 130-170: nowhere in this part is written that the development of the hypothalamic-pituitary-adrenal axis of the unborn child is fundamental for the upcoming parturition event. I think this should be added somewhere in this part.
lines 159-166 may be shortened and summarized.
- Line 235: please replace womb with uterus
- Line 440: ....depressive and long term adverse effects: which are these effects? please add here.
Author Response
Please, find enclosed the Review report of our manuscript. Thank you very much for your valuable suggestions and comments.

Reviewer 3 Report
In this study, the authors aimed to investigate and describe the effects of maternal distress in pregnancy and the postpartum period on adverse child developmental outcomes.
In general, the manuscript is well written. Data and discussion of the results are convincing, but some minor inconsistencies need to be clarified. I only have minor corrections and comments to the manuscript, which are outlined below.
Line 47-54: “The Developmental Origins of Health and Disease (DOHaD) hypothesis emphasizes the interdependence of developmental genetic or environmental influences and suggests that human health and development originate in early life”. Given the topic of the review, in my opinion, the understanding of the DOHaD paradigm is essential for the reader. I suggest expanding this section with specific examples and references.
Lines 216-223: The 11β-HSD2 is an essential placental enzyme correlated to the regulation of GCs concentrations. I suggest better explaining in this section the mechanism by which the enzyme modulates the GCs levels from maternal to fetal circulation.
Lines 233-238: add references in the figure 1 legends.
Lines 290-297: The authors stated “Currently, there is little evidence of direct or indirect interaction between maternal and fetal oxidative systems in human pregnancies”. This sentence at the end of the section looks ambiguous. Better explain the concept or delete it.
Lines 526-528: are there no more recent studies? Please check the literature.
Lines 555-556: which are these studies? Add references and report them.
Author Response
Please, find enclosed the Review report of our manuscript. Thank you very much for your valuable comments and suggestions.

Round 2
Reviewer 2 Report
Dear Authors,
All the changes requests and questions have been addressed correctly and exhaustively.
I just found one very small change to be done:
line 58: shows that fetuses whose mothers were malnutrition in the second trimester of pregnancy.... change the word "malnutrition" as it is not correct.
Author Response
Dear Reviewer,
Please find an enclosed review report of our manuscript (revision round 2).
Thank you very much for your valuable comments and suggestions.
